# No Evidence of Bacterial Symbionts Influencing Host Specificity in *Aphis gossypii* Glover (Hemiptera: Aphididae)

**DOI:** 10.3390/insects13050462

**Published:** 2022-05-14

**Authors:** Hao Guo, Fengying Yang, Min Meng, Jingjing Feng, Qinglan Yang, Yongmo Wang

**Affiliations:** Hubei Insect Resources Utilization and Sustainable Pest Management Key Laboratory, College of Plant Science & Technology, Huazhong Agricultural University, Wuhan 430070, China; haoguokuaby@163.com (H.G.); fengyingy953507@163.com (F.Y.); angrydp@hotmail.com (M.M.); jingjingf@webmail.hzau.edu.cn (J.F.); y1751118447@163.com (Q.Y.)

**Keywords:** *Aphis gossypii* Glover, host specificity, bacterial symbiont, *Buchnera* abundance, wing morphism

## Abstract

**Simple Summary:**

The cotton-melon aphid, *Aphis gossypii* Glover, is a polyphagous insect pest with many host-specialized biotypes. The mechanism of host specificity remains unknown in this aphid. In this study, we investigated whether bacterial symbionts control the host specificity of *A. gossypii*, as reported in some aphids. The two typical host-specialized biotypes used in this study produced significantly fewer nymphs on non-native hosts than on a native host, indicating a high host specificity of the two biotypes. We found that the winged morph of both biotypes had a significantly lower host specificity than its corresponding wingless morph. Bacterial analysis indicated that the composition of the bacterial symbionts was not different between the two biotypes, but within each biotype, the *Buchnera* abundance in the winged morph was only about 10% of that in the wingless morph. We suspected that a low *Buchnera* abundance was associated with a low host specificity. We compared the reproduction of *A. gossypii* with different *Buchnera* abundances, and did not find that a low *Buchnera* abundance resulted in a low host specificity. We then concluded that the host specificity of *A. gossypii* is not controlled by specific bacterial symbionts or *Buchnera* abundance.

**Abstract:**

The cotton-melon aphid, *Aphis gossypii* Glover, is a polyphagous insect pest with many host-specialized biotypes, such as the Cucurbitaceae- and Malvaceae-specialized (CU and MA) biotypes. Bacterial symbionts were reported to determine the host range in some aphids. Whether this is the case in *A. gossypii* remains unknown. Here, we tested the host specificity of the CU and MA biotypes, compared the host specificity between the wingless and winged morph within the same biotype, and analyzed the composition of the bacterial symbionts. The reproduction of the CU and MA biotypes reduced by 66.67% and 82.79%, respectively, on non-native hosts, compared with on native hosts. The composition of bacterial symbionts was not significantly different between the CU and MA biotypes, with a *Buchnera* abundance >95% in both biotypes. Meanwhile, the winged morph produced significantly more nymphs than the wingless morph on non-native hosts, and the *Buchnera* abundance in the winged morph was only about 10% of that in the wingless morph. There seemed to be a relationship between the *Buchnera* abundance and host specificity. We regulated the *Buchnera* abundance by temperature and antibiotics, but did not find that a low *Buchnera* abundance resulted in the high reproduction on non-native hosts. We conclude that the host specificity of *A. gossypii* is not controlled by specific bacterial symbionts or by *Buchnera* abundance.

## 1. Introduction

The cotton-melon aphid, *Aphis gossypii* Glover (Hemiptera: Aphididae), is a highly polyphagous and cosmopolitan insect pest that damages more than 600 horticultural and agricultural crops, such as cotton, okra, cucumber, eggplant, potato, and chrysanthemum [1]. Besides inflicting direct damage, it also inflicts indirect damage by transmitting plant viruses [2]. The lifecycle of *A. gossypii* is complex, typically including a holocycle and an anholocycle. A holocycle in *A. gossypii* was mainly reported in Asia and North America, characterized by an alteration between a sexual generation on primary host plants and multiple asexual generations on various secondary host plants in a year; an anholocycle in *A. gossypii* was mainly reported in Europe and Africa, characterized by continuous apomictic parthenogenesis on various secondary host plants [1]. *Aphis gossypii* displays a wing dimorphism in response to environment conditions, with a wingless morph under favorable conditions (high host quality and low population density) and a winged morph under unfavorable conditions (low host quality and high population density) [3].

*Aphis gossypii* consisted of host-associated populations that performed much better on their native host plants than on others. Those populations are often called host-specialized biotypes [4]. For instance, in Europe, *A. gossypii* from cucumber and chrysanthemum were unable to establish populations after having their host plants exchanged [5]; *Aphis gossypii* from cucumber and eggplant also did not survive on eggplant, cotton, and okra [6]. In Australia, *A. gossypii* from cotton did not survive on cucumber and pumpkin, and clones from pumpkin did not survive on hibiscus and cotton [7]. Similarly, *A. gossypii* from cotton or hibiscus in China did not establish populations on cucumber, and vice versa [8,9,10]. It is an interesting ecological phenomenon that *A. gossypii* have such obvious differentiation in host utilization. The studies on the host specificity of *A. gossypii* normally used wingless aphids as the materials [8,9,10,11,12]. Under natural conditions, winged adult aphids develop and transfer to new host plants when host plants become poor in nutrition quality. Therefore, it is necessary to compare the degree of host specificity between the winged and wingless within the same biotype of *A. gossypii*.

Aphids normally harbor both obligate (primary) and facultative (secondary) bacterial symbionts [13,14]. Host specificity was probably controlled by bacterial symbionts, which had been reported in pea aphid (*Acyrthosiphon pisum*) and cowpea aphid [15,16,17,18,19]. The obligate bacterial symbiont, *Buchnera*, is indispensable for aphids because it synthesizes nutrients that are deficient in the sap diet of aphids [20]. Facultative bacterial symbionts are not indispensable, but some of them perform special functions for their aphid hosts, such as heat tolerance [21,22], parasite or pathogen resistance [23,24], and host specificity [15]. Some researchers studied the diversity of bacterial symbionts of *A. gossypii* collected from different host plants. For instance, Xu et al. [25] reported that the host plants rather than the geography seemed to have shaped the symbiont composition of *A. gossypii*, Ma et al. [26] found that several facultative symbiotic bacteria were associated with specific host plants of *A. gossypii*, and Najar-Rodríguez et al. [27] reported that the symbiont composition of *A. gossypii* collected from Japan and Australia reflected the location more than the host plant. Whether host specificity is determined by specific bacterial symbionts or by an abundance of a specific bacterial symbiont remains to be studied in *A. gossypii* using stable laboratory populations of host-specialized biotypes.

We maintained asexual lines of the CU and MA biotypes in the laboratory. These lines showed a high host specificity on their native hosts [28]. This study aimed to compare the host specificity of winged and wingless *A. gossypii*, and to test the effects of endosymbionts on the host specificity of *A. gossypii*. We first compared the host specificity between the wingless and winged morphs of both biotypes, followed by a comparison of the endosymbionts’ composition between the CU and MA biotypes, and the *Buchnera* abundance between the wingless and winged morphs. Significant differences in the host specificity and the *Buchnera* abundance between the winged and wingless morphs were found, so we finally evaluated the effects of the *Buchnera* abundance on the host specificity of the two biotypes.

## 2. Materials and Methods

### 2.1. Aphid and Plant Materials

*Aphis gossypii* were collected from cucumber (*Cucumis sativus* L.) and hibiscus (*Hibiscus syriacus* L.) from Baoding (38°53′ N; 115°28′ E), China, in the spring of 2015. Aphids from cucumber and hibiscus were called Cucurbitaceae-associated (CU) biotype and Malvaceae-associated (MA) biotype, respectively. The aphids were maintained on cucumber (variety Lufeng) and cotton (variety Suzamian 3) plants in nylon net cages (0.16 mm mesh size) in an artificial climate chamber at 20 ± 2 °C with a 16L:8D photoperiod. To avoid overcrowding, aphids were transferred to fresh plants every two weeks. Cotton and cucumber plants were cultivated in pots (10 × 10 cm) with commercial nursery soil in a separate artificial climate chamber at 25 ± 2 °C with a 16L:8D photoperiod. Plant materials were used to culture aphids at the three- to six-leaf stages.

### 2.2. Host Specificity Comparison between Winged and Wingless Morph

In order to evaluate the influence of wing dimorphism on host specificity, we compared the performance between winged and wingless morphs of the two biotypes on non-native hosts. To obtain winged adults, more than 50 aphids of each biotype were introduced to one plant of their respective host in a nylon net cage (45 × 45 × 45 cm). Winged adults developed after two to three weeks due to overcrowding. Newly developed winged adults were collected twice a day from the inner walls of the nylon net cages. To obtain wingless adults, ten wingless adults were introduced to one plant of their respective hosts in a nylon net cage to give birth to nymphs and were removed 24 h later. The nymphs developed into wingless adults in six to seven days, and the newly developed wingless adults were used as wingless aphids. Winged and wingless adults of both biotypes were introduced to their own host and each other’s host, that were cultured as detached leaves in Petri dishes (Φ15 cm) filled with 20 mL of 1% agar gel. The Petri dishes were wrapped separately in transparent nylon net bags (0.16 mm mesh size) to prevent the aphids from escaping. Wingless and winged aphids of each biotype on their native host were used as controls. Each treatment included five replicates, each of whom consisted of one Petri dish with one detached leaf that were introduced with 10 aphids. Those aphids were cultured at 20 ± 2 °C with a 16L:8D photoperiod in a growth chamber. The number of aphids in each Petri dish was counted daily for 15 d.

### 2.3. Bacterial Symbionts Analysis of Host-Specialized Biotypes

Bacterial symbionts of CU and MA biotypes were analyzed by 16S rRNA gene sequencing. To obtain aphids of the same age, aphids of CU and MA biotypes were cultured on detached cucumber and cotton leaves in Petri dishes, respectively. Ten wingless adults were introduced into each Petri dish to give birth to nymphs and removed 24 h late. The nymphs were cultured in an artificial climate chamber at 20 ± 2 °C with a 16L:8D photoperiod, and developed into new wingless adults in 6 d. Three samples were prepared for each biotype and each sample consisted of ten wingless adults. The wingless adults were extracted for DNA using a universal DNA extraction kit (Takara, Kusatsu, Japan), according to the manufacturer’s protocol, after surface sterilization in 75% ethanol for 30 s. The final DNA concentration and purification were determined using a NanoDrop 2000 spectrophotometer (Thermo Scientific, Wilmington, NC, USA), and DNA quality was checked by running a 1% agarose gel electrophoresis. The V3-V4 hypervariable regions of the bacterial 16S rRNA gene were amplified with primers 341F (5′- CCTAYGGGRBGCASCAG -3′) and 806R (5′- GGACTACNNGGGTATCTAAT -3′) using the GeneAmp 9700 thermal cycler PCR system (Applied Biosystems, Foster City, CA, USA). The PCR reactions were carried out in a 50-μL volume containing 1.5 μL (10 μM) of each primer, 0.8 U DNA Polymerase (New England Biolabs, Ipswich, MA, USA), 1 μL dNTPs, and 40–60 ng DNA template. The resulting PCR products were extracted from a 2% agarose gel, further purified using the AxyPrep™ DNA Gel Extraction Kit (Axygen Biosciences, Union City, CA, USA), and quantified using QuantiFluor™-ST Fluorometer (Promega, Madison, WI, USA). Next, 10 μL of the purified product was ligated to adapter and sample barcode in a 50-μL volume containing 1 μL (10 μM) of each fusion primer and 20 μL of 2 × PCR Master Mix (New England Biolabs). The PCR conditions were as follows: 95 °C for 30 s, 10 cycles of 95 °C for 10 s, 65 °C for 30 s, and 72 °C for 30 s. The final PCR products were extracted from 2% agarose gel electrophoresis, and purified with VAHTS™ DNA Clean Beads (Vazyme Biotech), and then quantified by NanoDrop 2000. All positive PCR products were mixed at a mass ratio of 1:1. Finally, the Purified amplicons were submitted to an Illumina HiSeq 2500 platform (Illumina, San Diego, CA, USA) for paired-end sequencing. Raw FASTQ files were demultiplexed, quality-filtered using Trimmomatic, and merged using FLASH software (http://cbcb.umd.edu/software/flash (accessed on 1 May 2022)). The denoised sequences were clustered into operational taxonomic units (OTUs) at 97% similarity. Taxonomy was assigned to all OTUs by searching against the Silva databases. The OTUs were then filtered with a threshold value of 0.005% of all sequences. Finally, an OTU table containing the number of sequences per sample and taxonomic information was generated.

### 2.4. Buchnera Quantification of Winged and Wingless Aphids

The abundance of *Buchnera* in winged and wingless adults was quantified by both mycetocyte counting and quantitative PCR (qPCR) methods. Newly developed winged and wingless adults of CU and MA biotypes were dissected individually to count mycetocytes according to the method proposed by Cloutier and Douglas [29]. The adult aphids were fixed in modified Bouin–Dubosq solution for 24 h. The fixed aphids were washed in 75% ethanol and dissected in a drop of distilled water under binocular stereomicroscope (×40 magnification) with fine pins. Mycetocytes spread from aphid body cavity into distilled water and were counted. Each treatment consisted of 20 to 30 individuals. Total DNA was extracted from aphid materials using a universal DNA extraction kit for qPCR quantification. *Buchnera* abundance was quantified using EF1-α as an internal reference gene [30,31]. qPCR was performed using SYBR Premix Ex Taq (Takara, Japan) in a CFX96 Touch Real-Time PCR Detection System (Bio-Rad, Hercules, CA, USA). The 16S rRNA gene of *Buchnera* was amplified using the primer set Buch16S1F (5′-GAGCTTGCTCTCTTTGTCGGCAA-3′) and Buch16S1R (5′-CTTCTGCGGGTAACGTCACGA-3′), and the EF1-α gene of *A. gossypii* was amplified with the primer set EF1-αF (5′-TATGGTGGTTCAGTAGAGTC-3′) and EF1-αR (5′-CTGATTGTGCCGTGCTTATTG-3′). The total volume of the real-time qPCR reaction was 20 µL, containing 2 μL of forward and reverse primers (10 mM), 10 μL of SYBR Premix Ex Taq (Tli RNase H Plus, Takara), 7.0 μL of sterile water, and 1 μL of bacterial genomic DNA (<100 ng). The PCR cycling conditions were as follows: initial denaturation at 95 °C for 3 min, 34 cycles of 95 °C for 30 s, 55 °C for 30 s, 72 °C for 30 s, and 10 min at 72 °C at the end of the cycle. Each treatment consisted of three individuals and each one underwent three qPCR quantifications.

### 2.5. Evaluating the Effects of Buchnera Abundance on Host Specificity

We regulated *Buchnera* abundance by temperature and antibiotics because *Buchnera* are sensitive to temperature and antibiotics [32,33]. Adults of CU and MA biotypes were introduced to their respective native plants in the nylon net cage to give birth to nymphs for 24 h. Newborn nymphs were cultured at five constant temperatures of 15 °C, 20 °C, 25 °C, 30 °C, and 35 °C for one month. Host plants were replaced with intact plants every 10 d. The mycetocyte counting method was used to quantify *Buchnera* abundance under each temperature (30 measurements per temperature). The most suitable temperature for growth and reproduction of *A. gossypii* ranges from 18 °C to 30 °C [34,35,36,37,38]. In order to avoid the influence of extreme temperatures on aphid physiology, only the aphids cultured at 20 °C, 25 °C, and 30 °C were used in the following experiments. Ten newly developed wingless adults cultured at each temperature were transferred to native and non-native hosts in Petri dishes and cultured at 20 ± 2 °C in a growth chamber. Wingless CU and MA on their respective native hosts were used as controls. Each treatment consisted of five replicates (five Petri dishes with detached leaves). Reproduction of these aphids was recorded daily for 10 d.

A cocktail of antibiotics was prepared by mixing rifampicin, oxytetracycline hydrochloride, neomycin sulfate, and doxycycline in equal weights to a final concentration of 200 μg antibiotics per milliliter. Two milliliters of the cocktail were sealed within two layers of stretched parafilm membrane. Newly developed wingless adults of CU and MA biotypes were fed with the sealed cocktail at 20 ± 2 °C and a 16L:D8 photoperiod for 40 h. A diet of 3% sucrose solution with no antibiotics served as a control. Average abundance of *Buchnera* under each concentration was estimated also using the mycetocyte counting method. Newly developed wingless adults were fed with antibiotics for 40 h, were transferred to cotton and cucumber leaves at a density of 10 aphids per leaf in Petri dish, and then were cultured in a growth chamber at 20 ± 2 °C. Wingless CU and MA on their respective native hosts were used as controls. Each treatment consisted of five replicates (five Petri dishes with detached leaves). Reproduction of these aphids was recorded daily for 10 d.

### 2.6. Data Analysis

The number of OTUs at the genus level was compared between CU and MA biotypes of *A. gossypii* using the Wilcoxon rank-sum test. The overall effects of wing morphism on reproduction of host-specialized *A. gossypii* on non-native host plants were analyzed by repeated-measures ANOVA, and the overall means were compared using Tukey’s HSD test. Meanwhile, the daily reproduction of winged and wingless aphids on non-native hosts was compared using Tukey’s HSD in one-way ANOVA. *Buchnera* abundance was compared between winged and wingless CU and MA biotypes using Tukey’s HSD in one-way ANOVA. The effects of temperature and aphid biotype on *Buchnera* abundance were analyzed using two-way ANOVA, and means within each factor were compared using Tukey’s HSD. *Buchnera* abundance before and after antibiotic treatment was compared using an independent *t*-test. The effects of *Buchnera* abundance on the performance of CU and MA biotype on non-native hosts were also analyzed using repeated-measures ANOVA followed by Tukey’s HSD. Data were $\sqrt{x+1}$ transformed in ANOVA tests to meet the assumptions of normality and homogeneity of variance. Statistical analyses were performed using the IBM SPSS statistics package version 19.0 (IBM, Armonk, NY, USA).

## 3. Results

### 3.1. Host Specificity of Winged and Wingless Host-Specialized Biotypes

Compared with the reproduction of the wingless and winged CU biotype on cucumber, the reproduction of the wingless CU biotype on cotton decreased by 66.67% and 49.31%, respectively, during the 15-d culture (Figure 1A). Compared with the reproduction of the wingless and winged MA biotype on cotton, the reproduction of the wingless MA biotype on cucumber decreased by 82.79% and 68.16%, respectively (Figure 1B). The results indicate the host-specialized lines had a high host specificity to their respective native hosts. However, compared with the reproduction of the wingless CU biotype on cotton, the reproduction of the winged CU biotype on cotton increased by 51.52% (Figure 1A). Compared with the reproduction of the wingless MA biotype on cucumber, the reproduction of the winged MA biotype on cucumber increased by 103.59% (Figure 1B). Therefore, the winged morph performed much better than the wingless morph on non-native hosts.

Repeated-measures ANOVA revealed that the winged morph produced significantly more nymphs than the wingless morph in both the CU (*p* < 0.001, Tukey’s HSD) and MA biotype (*p* < 0.001, Tukey’s HSD) (Figure 1A,B). The daily population of the winged CU biotype was significantly greater than that of the wingless CU biotype on cotton from the 8th day to the end of the experiment (*p* < 0.050, Tukey’s HSD in one-way ANOVA) (Figure 1A), and that of the winged MA biotype was significantly greater than that of the wingless MA biotype on cucumber from the 5th day to the end of the experiment (*p* < 0.050, Tukey’s HSD in one-way ANOVA) (Figure 1B). These results show that winged aphids of both biotypes performed significantly better (had a lower host specificity) than their respective wingless morphs on non-native hosts. The nymphs of the CU biotype developed into largish green individuals on cotton, and the nymphs of the MA biotype developed into abnormal yellow dwarfs on cucumber (Figure 2).

### 3.2. Symbiont Composition in CU and MA Biotype

We obtained 287,085 reads across the six samples (47847 reads per sample on average). A total of 204 OTUs were identified at 97% similarity. The OTUs were assigned into 8 phyla, 19 classes, 34 orders, 53 families, and 64 genera. In both the CU and MA biotypes, the relative abundance of the obligate symbiont *Buchnera* was over 95%, and the most abundant facultative symbionts were below 3.06% (Figure 3A). *Microbacterium*, *Bradyrhizobium*, *Paenarthrobacter*, *Arsenophonus*, and *Allorhizobium* were the top five facultative symbionts according to average abundance across all samples, and their relative abundances were 0.51%, 0.25%, 0.24%, 0.19%, and 0.11%, respectively. A separate analysis without *Buchnera* reads indicated that none of facultative symbionts were statistically different in abundance between the two biotypes (*p* > 0.050, Wilcoxon rank-sum test), and no facultative symbiont was exclusively associated with the CU or MA biotype (Figure 3B). The abundance of *Microbacterium* was very high in one of the CU samples, but not in the other two CU samples. So, facultative symbionts were unlikely to control host specificity in our laboratory populations of the CU and MA biotypes.

### 3.3. Buchnera Abundance on Winged and Wingless Morphs

The winged morph harbored only approximately 10% as many mycetocytes as the wingless morph in both the CU and MA biotype, and the differences were significant (*p* < 0.001, Tukey’s HSD) (Figure 4A). Similarly, the relative copy number of *Buchnera* in the winged aphids was approximately 5% that of the wingless aphids in both biotypes, and the differences were also significant (*p* < 0.001, Tukey’s HSD) (Figure 4B). The abundance of *Buchnera* in the CU biotype was also found to be significantly higher than that in the MA biotype, both in terms of the mycetocyte number (*p* < 0.031, Tukey’s HSD) and the relative copies of *Buchnera* (*p* < 0.015, Tukey’s HSD) (Figure 4A,B).

### 3.4. Response of Buchnera Abundance to Temperature and Antibiotics

One-way ANOVA showed that temperature had a significant effect on the *Buchnera* abundance in both the CU biotype (*df* = 4.42; *F* = 112.062; *p* < 0.001) and the MA biotype (*df* = 4.42; *F* =67.415; *p* < 0.001). The *Buchnera* abundance was highest at 25 °C, and the *Buchnera* abundance at 35 °C was only about 10% of that at 25 °C (Table 1). Two-way ANOVA showed that both the temperature (*df* = 4.76; *F* = 163.92; *p* < 0.001) and host biotype (*df* = 1.76; *F* = 16.83; *p* < 0.025) significantly affected the *Buchnera* abundance. Again, the *Buchnera* abundance of the CU biotype was found to be significantly higher than that of the MA biotype (*df* = 1.76; *F* = 16.83; *p* < 0.025) (Table 1).

After 40 h of feeding on the antibiotic cocktail diet, the *Buchnera* abundance of the CU and MA biotypes decreased by approximately 80%, with a significant difference between the antibiotic and antibiotic-free treatments (*p* < 0.001, independent *t*-test) (Table 1).

### 3.5. Reproduction of Aphids with Different Abundance of Buchnera on Non-Native Hosts

Repeated-measures ANOVA indicated that the *Buchnera* abundance regulated by temperature had significant effects on the reproduction of the CU biotype on cotton (*df* = 2.12; *F* = 8.176; *p* = 0.006) (Figure 5A) and of the MA biotype on cucumber (*df* = 2.12; *F* = 10.804; *p* = 0.002) (Figure 5B). The CU and MA biotypes with a high *Buchnera* abundance (regulated at 25 °C) produced significantly more nymphs than the same biotype with a low *Buchnera* abundance (regulated at 30 °C) on non-native host plants (*p* = 0.004 and *p* = 0.003, Tukey’s HSD) (Figure 5A,B). There was no significant difference in reproduction between the median Buchnera abundance (regulated at 20 °C) and the high *Buchnera* abundance (regulated at 25 °C) in non-native host plants in both the CU biotype (*p* = 0.186, Tukey’s HSD) and the MA biotype (*p* = 0.925, Tukey’s HSD) (Figure 5A,B).

The CU biotype with a high *Buchnera* abundance (antibiotic-free) produced significantly more nymphs than the CU biotype with a low *Buchnera* abundance (antibiotic-treated) (*df* = 1.8; *F* = 34.964; *p* < 0.001; repeated-measures ANOVA) (Figure 5C). The MA biotype with a high *Buchnera* abundance (antibiotic-free) also produced significantly more nymphs than the MA biotype with a low *Buchnera* abundance (antibiotic-treated) (*df* = 1.8; *F* = 61.559; *p* < 0.001; repeated-measures ANOVA) (Figure 5D).

These results show that the performance of host-specialized *A. gossypii* on non-native hosts was not negatively correlated with the *Buchnera* abundance as we had expected.

## 4. Discussion

The host specificity tests indicated that the aphid materials used in this study were highly specialized on their native hosts. The CU and MA biotypes maintained a low level of population growth on the non-native hosts (Figure 1), which are different from the results of previous studies [7,8,9,10] in which the MA and CU biotypes died within 5–7 days when transferred to non-native host plants. This discrepancy may be caused by the temperature settings. Our experiments were carried out at 20 ± 2 °C, while the previous studies were carried out at 25 °C to 28 °C. When the temperature is higher than 25 °C, the fecundity of *A. gossypii* is inhibited [39]. In addition, host-specialized biotypes from different regions may differ in the degree of host specificity [40].

Previous studies used wingless morphs of the CU and MA biotypes to test the host specificity of *A. gossypii* [6,9,10]. Under natural conditions, aphids normally transfer host plants as winged morphs. In this study, we specifically compared the performance of winged and wingless morphs of the CU and MA biotypes on non-native host plants. We found that the reproduction of the winged CU or MA biotypes was significantly higher than that of their corresponding wingless morphs on non-native hosts (Figure 1), indicating that the winged morph of both biotypes had a lower host specificity than their wingless morph. This is the first time that significant differences in host specificity have been found between winged and wingless morphs of the same biotype of *A. gossypii*. The low host specificity of the winged morph indicated that host-specialized biotypes of *A. gossypii* can move as a winged morph to survive for a short period on non-native hosts during the senescence of the native hosts. Although we found that the winged morph produced significantly more nymphs than their corresponding wingless morph on non-native host plants, it did not mean that the winged morph could fully adapt to the non-native hosts. The body size and body color of the aphids were apparently abnormal in the non-native hosts, especially the MA biotype on cucumber (Figure 2).

The abundance of the obligate symbiont *Buchnera* was more than 95% in both the CU and MA biotype of *A. gossypii*. The average abundances of the facultative symbionts, however, including the top five genera, *Microbacterium*, *Bradyrhizobium*, *Paenarthrobacter*, *Arsenophonus*, and *Allorhizobium*, were below 0.51%. Ma et al. [26] analyzed symbiotic bacteria of *A. gossypii* collected from five host plants, and they found that the most abundant was *Buchnera* (ranged from 52.61% to 93.54%), followed by *Serratia* (ranged from 0.00% to 23.64%) and *Arsenophonus* (ranged from 0.01% to 1.99%). Xu et al. [25] found that the bacterial community of *A. gossypii* collected from 25 host plants was dominated by *Buchnera* (average abundance 91.79%), followed by *Arsenophonus* (1.11%) and *Acinetobacter* (0.99%). In contrast, our aphid samples had the lowest microbial diversity, with the most abundant species being *Buchnera* (>95%). Our samples were reared in a laboratory, while theirs were collected in nature, which could account for the difference. We found that the facultative symbionts were not distributed differently between the CU and MA biotypes, and no specific facultative symbiont was exclusively associated with a specific host biotype. Therefore, the host specificity, at least in our samples of *A. gossypii*, was not controlled by facultative symbionts, as reported in other aphid species, such as the pea aphid *A. pisum* [15,16].

We quantified the *Buchnera* abundance in winged and wingless aphids using both the mycetocyte counting method and the quantitative PCR method, and found that the *Buchnera* abundance in winged aphids was only approximately 10% of that in wingless aphids. The differences in the *Buchnera* abundance between winged and wingless aphids were consistent with previous studies on other aphid species. For instance, the numbers of mycetocytes in wingless *A. pisum* and *Aphis fabae* were significantly higher than that in the corresponding winged morph [41,42]. The *Buchnera* abundance of winged *A. gossypii* was lower than that of the wingless morph, and the degree of host specificity of the winged morph was lower than that of the wingless morph. We speculated that there was a negative correlation between the *Buchnera* abundance and host specificity. The reasons for this speculation are: There was a strict symbiotic relationship between aphids and *Buchnera* [13,14]; when the wingless host-specialized aphids are transferred to unfavorable hosts, *Buchnera* does not immediately reduce in quantity and nutrient requirements; *Buchnera’s* nutritional needs stimulate the aphids to ingest phloem sap on unfavorable hosts; however, the aphids cannot effectively metabolize the harmful secondary substances in phloem sap, so their performance on non-native hosts become poor. In contrast, the winged morph has a low *Buchnera* abundance and low phloem sap ingestion, so the winged aphids performed better than the wingless aphids on non-native hosts. Subsequently, we tested our speculation.

We regulated the *Buchnera* abundance by temperature and antibiotics. The *Buchnera* abundance of the CU and MA biotype of *A. gossypii* reached its peak at 25 °C. When the temperature exceeded 30 °C, the *Buchnera* abundance decreased sharply to 10% of its peak level. Aphid body size and color became abnormal at high temperatures (30 °C and 35 °C). The results are similar to those reported in other aphids. For instance, Chen et al. [31] reported that the number of mycetocytes of *A. craccivora* increased significantly with the development of the aphids at suitable temperatures of 15 °C and 20 °C, and decreased significantly with the development of the aphids at high temperatures of 30 °C and 35°C. The abundance of *Buchnera* on *A. pisum* decreases with temperature and age [33]. Therefore, the abundance of Buchnera can be regulated by temperature.

The abundance of *Buchnera* on *A. gossypii* decreased by approximately 80% after 40 h of antibiotic treatment. Rifampin and ampicillin completely removed the facultative symbiont Rickettsia in *Bemisia tabaci* [43], and rifampin also completely removed *Wolbachia* from *Echinothrips americanus* [44]. Similarly, the combination of ampicillin, cephalosporin, and gentamycin sulfate antibiotics completely removed the secondary symbiont *Regiella insecticola* in *Sitobion avenae* [45,46]. We tested different concentrations of oxytetracycline hydrochloride, rifampicin, and neomycin sulfate to remove the obligate symbionts *Buchnera* in *A. gossypii*, but only 20% of *Buchnera* was removed. Therefore, we added doxycycline to produce a cocktail containing four antibiotics. After 40 h of feeding on this cocktail, the number of mycetocytes in aphids decreased by 70–80%. After 40 h of antibiotic treatment, the aphids were transferred to leaves soaked with antibiotics for further cultivation for 15 to 20 days, and we found that the *Buchnera* abundance did not decrease further, which indicated that feeding aphids with mixed antibiotics for 40 h was effective to remove endosymbionts.

We transferred the host-specialized *A. gossypii* with different *Buchnera* abundances regulated by temperature and antibiotics to non-native hosts. Contrary to our expectation, aphids of neither the CU nor MA biotypes with a low *Buchnera* abundance performed better on non-native hosts than those with a high abundance of *Buchnera*; on the contrary, the performance of the former was worse than that of the latter, showing high mortality and low reproduction (Figure 5). So, we conclude that the better performance of the winged *A. gossypii* on non-native host plants was not caused by a low *Buchnera* abundance. Except for bacterial symbionts, other factors may cause the decreased host specificity of the winged morph. First, winged aphids may acquire a strong tolerance to adverse conditions during their development into the winged morph. Second, winged aphids can deploy their stored energy materials to maintain reproduction on unfavorable hosts, while wingless aphids can only use energy indirectly from feeding to support reproduction.

## 5. Conclusions

This study did not find differences in the bacterial symbiont composition between the CU and MA biotypes and did not detect any facultative symbionts that were strictly associated with specific host-specialized biotypes. The host specificity of *A. gossypii* appears not to be controlled by facultative symbionts. The performance of the winged *A. gossypii* was significantly better than that of the corresponding wingless aphids on non-native hosts, indicating that host-specialized *A. gossypii* can be transferred to non-native hosts by winged aphids to temporarily survive during periods when there are no native host plants in the field. The abundance of *Buchnera* in winged aphids was only 10% of that in wingless aphids, but we did not find that *A. gossypii* with a high *Buchnera* abundance performed better on non-native hosts. Therefore, there is no evidence of the bacterial symbiont composition or *Buchnera* abundance controlling the host specificity in *A. gossypii*. We suggest that the mechanism of the host specificity of *A. gossypii* from factors other than bacterial symbionts should be studied in the future.

## Figures and Tables

**Figure 1 insects-13-00462-f001:**
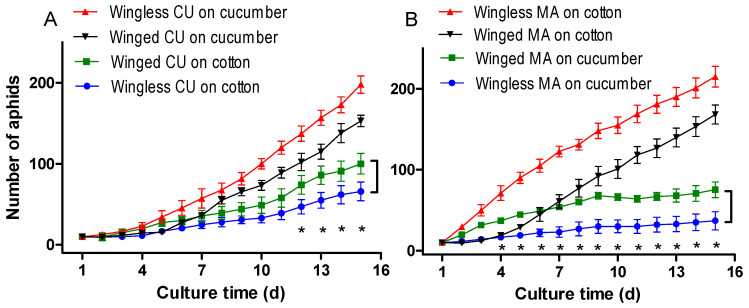
Cumulative reproduction of winged and wingless CU and MA biotypes of *A. gossypii* on non-native hosts. (**A**), wingless and winged CU on cotton, wingless and winged CU on cucumber acting as controls; (**B**), wingless and winged MA on cucumber, wingless and winged MA on cotton acting as controls. Asterisk * indicates significant difference of cumulative reproduction between winged and wingless aphids on non-native hosts at *p* = 0.05 (Tukey’s HSD in one-way ANOVA).

**Figure 2 insects-13-00462-f002:**
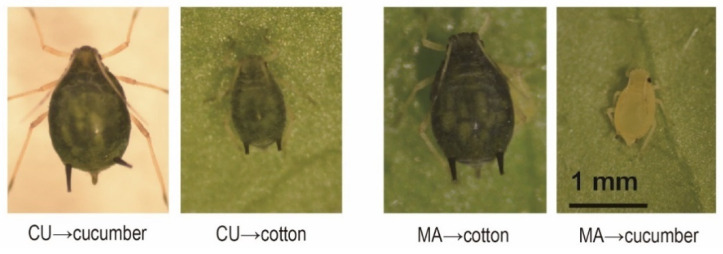
Morphology of Cucurbitaceae- (CU) and Malvaceae-specialized (MA) biotype of *A. gossypii* developed in native and non-native hosts.

**Figure 3 insects-13-00462-f003:**
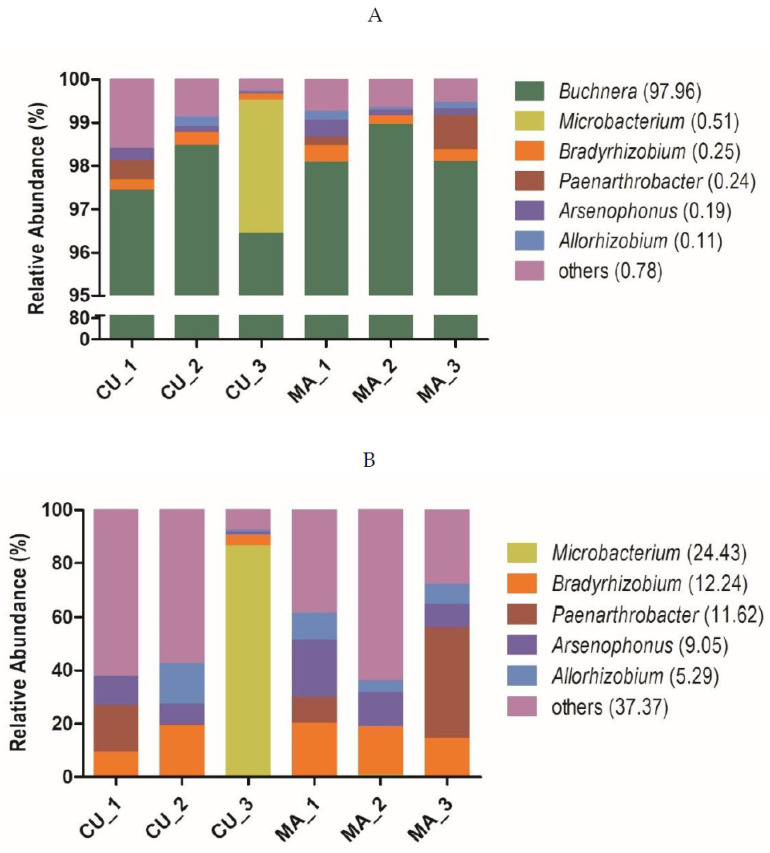
Relative bacterial symbiont abundance at the genus level in wingless CU and MA biotype of *A**. gossypii*. (**A**) *Buchnera* was included; (**B**) *Buchnera* was excluded. Each biotype consisted of three replications. The top six operational taxonomic units (OTUs) were listed and the rest of the OTUs were assigned to ‘others’. The number in brackets after the genus name indicates percentage across all samples. None of the OTUs were significantly different in abundance between CU and MA at *p* = 0.05 (Wilcoxon rank-sum test).

**Figure 4 insects-13-00462-f004:**
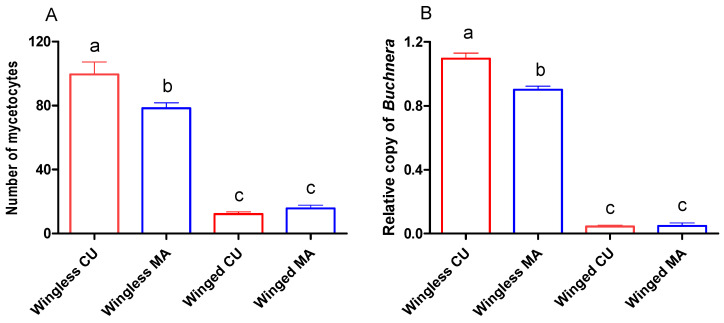
*Buchnera* abundance of winged and wingless CU and MA biotypes of *A. gossypii* quantified by mycetocyte counting method (**A**) and quantitative PCR method (**B**). Error bars indicate standard error. The same letters indicate no significant difference at *p* = 0.05 (Tukey’s HSD in one-way ANOVA).

**Figure 5 insects-13-00462-f005:**
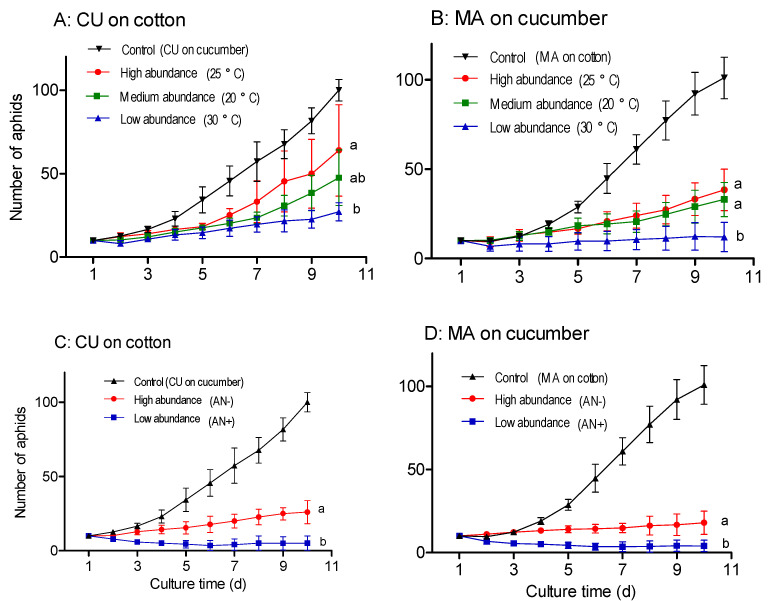
Effects of *Buchnera* abundance that was regulated by temperature and antibiotics on reproduction of host-specialized *A. gossypii* on non-native host plants. *Buchnera* abundance was regulated by temperature (**A**,**B**) and antibiotics (**C**,**D**). Number of aphids indicated cumulative reproduction. Error bars indicate the standard deviation. *Buchnera* abundance: 25 °C > 20 °C > 30 °C, AN- > AN+ (see Table 1). The controls acted as baseline and were not included in statistics. The same letters at the end of population growth-curves indicated no significant effects of *Buchnera* abundance on reproduction at *p* = 0.05 (repeated-measures ANOVA followed by Tukey’s HSD).

**Table 1 insects-13-00462-t001:** Response of *Buchnera* abundance of *A. gossypii* to temperature and antibiotics.

Factor	Level	CU	MA
Temperature			
	15 °C	49.00 ± 3.29 b	35.22 ± 2.24 c
	20 °C	61.22 ± 6.12 b	55.70 ± 4.98 b
	25 °C	108.10 ± 5.02 a	79.00 ± 2.61 a
	30 °C	25.70 ± 2.69 c	22.70 ± 5.51 c
	35 °C	10.50 ± 2.58 d	5.20 ± 1.25 d
Antibiotics			
	−(0 μg/mL)	102.08 ± 2.62 a	74.00 ± 2.52 a
	+(200 μg/mL)	17.58 ± 3.65 b	14.24 ± 2.51 b

Note: Buchnera abundance was expressed by number of mycetocytes per aphid. Values (mean ± standard error) followed by different letters within the same column of the same factor were not significantly different at *p* = 0.05 (one-way ANOVA followed by Tukey’s HSD for effect of temperature, and independent *t*-test for effect of antibiotics).

## Data Availability

Data are available via the Dryad Digital Repository at https://doi:10.5061/dryad.fxpnvx0v9 (accessed on 1 May 2022).

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
