# Peer review of "No Evidence of Bacterial Symbionts Influencing Host Specificity in Aphis gossypii Glover (Hemiptera: Aphididae)"

_insects, 2022, doi:10.3390/insects13050462_

Round 1

Reviewer 1 Report

The authors of the manuscript sought to investigate whether host specificity across two biotypes (and across two morphs from both biotypes) of the cotton aphid was underscored by differences in the abundances of the obligate and facultative symbionts, as well as other parameters. This is an interesting paper as the main rationale i could decipher was to see if this same host specific induced differences was similar to the pea aphid and the cowpea aphid. The submission is sound and I have a couple of suggestion to make to make it a more solid submission.

  1. Line 27: Add respectively after 82.7%
  2. Linea 231-240: Since Buchnera is an obligate endosymbiont, one would expect to find it in relatively high abundances in aphids. This has been established in several aphid lineages. I suggest that, perhaps remove all Buchnera reads from your 16S miseq data and do a separate analyses without it, and present the result for only the facultative bacteria in a separate figure (say Fig. 3B). This might make comparing or talking about the facultative symbionts more compelling.
  3. Also provide a table of the pairwise comparisons of the relative abundances among the facultative symbionts.
  4. Discussion.  Perhaps the authors can  discuss a bit more the ecological and evolutionary  relevance of the differences between the winged and wingless morphs across both the CU and  MA biotypes. Why does the winged have lower Buchnera abundances and also lower host specificity, while the wingless is the reverse?

Reviewer 2 Report

The manuscript (insects: 1693024) by Guo et al examines the host specificity of winged and wingless morphs of two biotypes (CU and MA) of Melon-cotton aphid Aphis gossypii. It analyzed the microbiome of the two biotypes and also examines the effect of the Buchnera abundance on the host specificity was assessed. The results showed that winged morphs of both biotypes have lower host specificity than wingless ones. The low Buchnera abundance regulated by temperature and antibiotics did not result in high reproduction on non-host species. Thus, the authors concluded that the host specificity of A. gossypii is not controlled by Buchnera abundance. Overall, the paper is interesting, but I find that some key data is missing to draw the above conclusions. Also some sections of the manuscripts especially methods and figures need to be improved. Below are my suggestions/comments for the authors to consider.

My major concerns are:

  1. Based on the performance and low Buchnera density in the winged morphs on the non-native host and by the regulated Buchnera experiment, the authors concluded that the host specificity is not controlled by bacterial symbionts or by Buchnera I am not convinced that the data provided is sufficient to make such conclusions for the following reasons: 

a). The data generated in this study is from one generation and I think it needs to be extended for a few more generations to see if the pattern holds the same. 

b). I think some key data is missing to interpret the results completely. For eg: The reproduction data of winged forms on their natural host is missing in Figure 1? Similarly, the reproduction data related to the effect of temperature and antibiotics (Buchnera abundance) on winged aphids on their natural host was not reported (Figure 5). This is important data because it will act as control/baseline of these insects after being exposed to varying temperatures and high doses of the antibiotic cocktail. 

c). Some of the arguments are not very clear and even confusing. For eg: “Line 279 – 280: These results showed that the performance of host-specialized gossypii on non-native hosts was not negatively correlated with Buchnera abundance as we had expected.” But it is not clear why this is expected? It would be very helpful if the authors well defined the hypothesis in the introduction.

  1. Microbiome and Buchnera Analysis: One of the main aims of the paper was to compare the wingless and winged morphs of the two biotypes,  but why is the microbiome analysis limited to wingless morphs only? The comment extends to Buchnera’s abundance as well. It is quantified on the natural host only. It is not clear how the non-native host plant and its metabolites alter the total microbiome and the Buchnera abundance in the winged and wingless morphs in the two biotypes. Also, the methodology as written is very incomplete for microbiome analysis. No information was given about the multiplexing of the samples. The authors need to include the sequence information (eg: total sequences generated, number of sequences used for analysis etc.)  in the results. Also please include the accession number and public database where the sequences were submitted.
  2. Mycetocyte counting method: This method is widely used in the paper to quantify/estimate the Buchnera abundance in the paper, yet the method is never described in the paper. Moreover, the citation “29” referred (Line 148) doesn’t have any information about the methodology. I suggest the authors to include the mycetocytes counting methodology in the paper and provide the correct citation.

 Other Comments:

Line 17-19: Sentence is not clear, please rephrase

Line 55 – 56: This statement doesn’t sound correct. Please rephrase.

Line 73: edit to “studied the diversity of bacterial symbionts”

Line 84: correct to “host specificity of winged and wingless”

Figures 1A and 1B: Figure legend: correct spelling of “Winged”

Figure 1: It is not clear if the data is mean population/day or cumulative means. Please clarify in the figure legend.

Round 2

Reviewer 2 Report

I have a few comments for the authors to consider below:

Since the authors cannot provide multi-generation data and considering the variation in the Buchnera abundance and performance of winged and wingless individuals on the non-native host, I strongly suggest including the reproduction data of winged individuals on native hosts in Figure 1. Similarly, the reproduction data of wingless biotypes (Buchnera abundance on fitness cost) on the native host should be added to figure 5 (Line 170-182) and the relevant text to be included in the results, which will strengthen the results and the paper.   

Microbiome methods: No additional information was added to the microbiome analysis. Line 140-141: Purified amplicons were pooled in equimolar quantities and paired-end sequenced….”.  There is confusion at this part. When are the samples multiplexed? Also, the microbiome data analysis part is too concise. Please add more details regarding sequence analysis. Considering the complexity of microbiome data, it is not possible to justify fitting all the information in 2 figures. Thus, I strongly suggest authors add the microbiome data to the public database and provide the details in the paper in case anyone is interested to look further.

Line 236: It is unusual that all the samples have the same read number. Does the number means, the reads that were analyzed or obtained?  To avoid confusion, I suggest the authors provide the read data as an excel file in the supplementary section.

Fig:1: Legend still didn’t clarify if it is cumulative or daily population. Please add these details.

Scientific names were not italicized in several places. Eg: Serratia (Line 358); Acinetobacer (Line 361). Please correct similar errors throughout the manuscripts.

Round 3

Reviewer 2 Report

Just a minor comment.

The provided Dryad link is not working and I could not access the data. Please make sure it works.